# A ‘Cultural Models’ Approach to Psychotherapy for Refugees and Asylum Seekers: A Case Study from the UK

**DOI:** 10.3390/ijerph21050650

**Published:** 2024-05-20

**Authors:** Mohaddeseh Ziyachi, Brian Castellani

**Affiliations:** 1Department of Sociology, Durham University, Stockton Road, Durham DH1 3LE, UK; brian.c.castellani@durham.ac.uk; 2Wolfson Research Institute for Health and Wellbeing, Durham University, Stockton Road, Durham DH1 3LE, UK; 3Durham Research Methods Centre, Durham University, Stockton Road, Durham DH1 3LE, UK; 4Centre for the Evaluation of Complexity Across the Nexus, University of Surrey, Guildford GU2 7XH, UK

**Keywords:** cognitive anthropology, ‘cultural models’ approach, migrants, refugees, asylum seekers, mental health, mental health treatment

## Abstract

Despite the existence of significant research on the mental health care challenges of migrants, particularly refugees and asylum seekers, less attention has been paid to treatment approaches. We used a case study from the UK to look at the topic from a cultural models approach (which comes from cognitive anthropology) to analyse migrants’ experiences with mental health care. Twenty-five refugees and asylum seekers living in North East England and Northern Ireland were interviewed who had used at least six sessions of talking therapy during the last three years. Our results suggested that adopting a ‘cultural models’ approach, which offers a new conceptual and methodological framework of migrants’ experiences and their underlying schemas and expectations, would significantly contribute to building therapeutic alliances and provide relevant and appropriate treatments for migrant clients, particularly for unrecognised pre- and post-migration traumatic experiences.

## 1. Introduction

Refugees and asylum seekers experience a higher rate of mental illness around the world [1,2,3,4]. A considerable number of studies have investigated the accessibility and effectiveness of mental health services for migrants to identify potential barriers and challenges [5]. Despite the existence of significant research on the mental health care challenges of immigrants, particularly refugees and asylum seekers, less attention has been paid to the treatment approach. This article uses a case study from the UK to look at the topic from a more fundamental perspective and analyse migrants’ experiences with mental health care in terms of the treatment approach. This research advocates for adopting a ‘cultural models’ approach in mental health treatment, which would contribute to building therapeutic alliances and providing relevant and appropriate treatments for migrant clients, particularly for unrecognised pre- and post-migration traumatic experiences.

Our paper is organised as follows. We first review the literature to characterise the problematic status of refugees’ and asylum-seekers’ mental health, particularly in terms of pre- and post-migration traumatic experiences. From there, we delineate our theoretical approach and methodology as being grounded in a ‘cultural models’ framework, as found in the field of cognitive anthropology. Finally, by reflecting on ethnographic data, we will discuss how a ‘cultural models’ approach, theoretically and methodologically, would contribute to obtaining insights into the migrant clients’ assumptions and expectations in mental health treatment and offering relevant and helpful therapies not just in the UK but in similar countries throughout the world. Finally, we must give a note on terminology. In this article, we will use ‘migrant’ as an umbrella term for refugees and asylum seekers. In turn, by ‘refugees’, we mean migrants who have received positive decisions for their asylum claims, and by ‘asylum seekers’, we mean those who are not yet legally recognised as refugees.

### 1.1. Migrants’ Mental Health and Its Causes

Non-stop human war, violence, and oppression have resulted in a continuous increase in forced displacement and migration, with 108.4 million refugees and asylum seekers (https://www.unhcr.org/global-trends, accessed on 1 March 2024). According to a systematic review of 26 studies conducted on 5143 adult refugees and asylum seekers from the Middle East, Europe, Asia, and 15 countries in Africa, refugees and asylum seekers experienced higher rates of mental illness, especially post-traumatic stress disorder (31.5%) and depression (31.5%), compared with 3.9% for PTSD and 12% for any depressive disorder among the general population [6]. Another systematic review of 40 studies with 11,053 participants indicated that the primary significant mental disorder observed among refugees and asylum seekers is major depressive disorder (MDD), with a prevalence rate of 32%. This was closely followed by post-traumatic stress disorder (PTSD) at 31%, recurrent episodes of MDD at 16%, and borderline personality disorder (BPD) at 5%, while the reported global prevalence for MDD is 4.4%, that for PTSD is between 3.9% and 5.6%, and 0.4% and 2.5% are the rates for psychosis and BPD, respectively [7].

Some studies have investigated the causes of higher rates of mental disorders for refugees and asylum seekers, including pre-migration traumas and post-migration stressors [8]. War and political- and state-based repression in the country of origin along with individual and institutional violence in the countries of transit are the most common pre-migration traumas in migrant populations [9]. Post-migration stressors encompass a range of challenges and restrictive policies, including uncertainty surrounding asylum-seeking applications and the risk of repatriation, lack of work permits, hardships in access to health, education, and other social services, and separation from family [10] (p. 23). A study conducted on treatment outcome questionnaires for 688 refugee and asylum-seeker patients in a mental health clinic in the Netherlands indicated that ‘lack of refugee status’ and ‘cumulative traumatic events’ are associated with the severity of PTSD symptoms and depression [8].

Recognition of the specific mental health conditions of refugees and asylum-seekers has contributed to the development of a movement towards providing mental health services in developed countries [11,12,13]. Consequently, a plethora of studies have focussed on mental health services used by migrants and the potential barriers and challenges that might keep them from using available services [5,14]. In the following section, some of these studies will be reviewed.

### 1.2. Using Mental Health Services

According to the available literature, practical barriers are the first category of challenges that keep refugees and asylum seekers from using mental health services. For instance, a systematic review by Byrow and colleagues [5] of 40 qualitative and 26 quantitative studies provided a list of key barriers. Their review indicated that ‘financial challenges’, ‘transport difficulty’, ‘unsecure housing’, ‘difficulty locating relevant services’, ‘appointment scheduling difficulty or delay’, ‘immigration status’, ‘continuity of care’, language barriers and ‘the lack of professional interpreters’, and limited knowledge about the healthcare system and services are the most commonly endorsed practical challenges in accessing mental health services.

In addition to practical barriers, negative attitudes and stigmatisations towards using mental health services and mistrust in mental health professionals keep asylum seekers and refugees from, for example, attending therapy sessions or cause them to drop out of treatment. For example, Schlechter and colleagues, [14] in their study on refugees and residents in Germany showed that refugees rejected the need for psychological treatment more than residents (p. 233). Byrow and collaegues, [5]) in their systematic review of 66 studies, also showed that mental health issues are stigmatised and imply negative connotations such as craziness and abnormality, bringing about negative consequences such as social shame, disapproval by family, or fear of being discriminated against in the community.

Other studies highlight how non-Western migrants regularly have a different understanding of mental health, its symptoms, and its causes, which may lead to a refusal to obtain information about mental health services, seek treatment, or utilise available services [14,15,16,17]. For example, Byrow and colleagues [5] found that in 18 reviewed studies, the participants employed somatic and physical terms such as ‘pain’ to describe their emotional conditions. They also found that while many migrants identified pre-migration traumas and post-migration experiences such as bereavement, uncertain refugee status, and discrimination as the causes of mental distress, some respondents associated it with religious or supernatural factors.

Similar challenges were found in terms of the perception of migrant and refugee participants towards the effectiveness of mental health treatments. The research shows a range of positive and negative beliefs about the benefits of professional treatments, including psychotherapy and medication [18,19,20,21]. Ellis and colleagues, [22] in their study on Somali refugee adolescents and their caregivers living in the northeast of the United States showed that mental health service providers, except for school counsellors, were not culturally or normatively accepted and familiar to the participants, especially those of the older generation. They would rather seek help through religion, such as repeating some parts of the Quran for the person with mental health, talking to family and community members or friends, and seeking advice from them. Byrow and colleagues [5] found that the participants in 10 reviewed studies expressed some concerns about the appropriateness or effectiveness of available mental health services because they assumed that available treatments were not culturally relevant to them, and service providers would not be able to understand their problems due to the cultural gap.

Other studies indicate that migrant clients regularly find talking therapy and assessment questions meaningless and ineffective for improvement [23,24]. That is why some studies posit that standardised treatment methods might not be culturally appropriate for non-Western clients, and they should be customised [25,26]. Instead, they emphasised adopting culturally sensitive treatments and tools [14,27,28] and employing a combination of treatment techniques for better outcomes [29].

## 2. The Purpose of the Current Research

As discussed above, the current literature shows that refugees and asylum seekers with specific life circumstances and traumatic experiences might find available mental health care irrelevant or unhelpful. Nevertheless, these studies do not address the impact of the therapeutic approach, nor do they propose what relevant treatments could be identified and adopted for migrant clients or how to identify them. The current article discusses how employing a ‘cultural models’ approach in mental health treatment would contribute to discovering the underlying assumptions and expectations of clients to provide relevant and effective treatments for them.

## 3. Historical Review of the Literature

In the following section, we provide a historical review of the literature on the links between psychotherapy, culture, and social context. From there, we use the work of Bennett [30] to explore the value of a cognitive anthropology framework for approaching psychological treatment. We end by introducing our ‘cultural models’ approach and its contribution to psychological treatment.

### 3.1. Psychotherapy, Culture, and Social Context

In response to the early directions in psychology, which were predominantly acultural, experimental, individual-based, and associated with scientific racism—and as a result of increasing the multicultural population in the US—*multicultural psychology* emerged in the second half of the 20th century [31]. Multicultural psychology criticises white-dominant approaches that do not consider the sociocultural backgrounds of clients. It emphasises considering the role of culture in therapy and its impacts on one’s lived experience, worldview, and behaviour and adopting a culturally relevant and appropriate treatment approach. It encourages therapists to obtain multicultural competency, which is the ‘cultural awareness, knowledge, and skills to effectively work with diverse clients’ [32] (p. 27).

Multi-cultural psychology’s term ‘cultural competency’ was also criticised later. It was suggested that mental health professionals needed to move beyond the medical model’s view of ‘competency’, as if one could master all cultures, to focus instead on ‘cultural humility, responsiveness, and reflexivity’ [30] (p. 286). As Kumagai and Lypson [33] explained, cultural competency is usually understood as a ‘static outcome’ which is actualised through obtaining ‘knowledge of characteristics, cultural beliefs, and practices of different non-majority groups, and skills and attitudes of empathy and compassion in interviewing and communicating with non-majority groups’ (p. 783). It also ignores the need for a continuous ‘critical consciousness’ of inequalities (p. 783).

Such critiques, including the need to attend to the cultural and social contexts of migrant mental problems and systematic inequalities and oppressions, then contributed to the emergence of the *social justice approach*. While multiculturalism psychology is office-based and not activity-based for changing structures, the social justice approach called for integrating advocacy into the practice to address sociopolitical factors and power relations that impact oppressed groups’ well-being and to try to change structural and systematic inequalities [32] (pp. 10–11). Both the multiculturalism and social justice approaches underline considering cultural and sociopolitical contexts in counselling. They invite therapists not to adopt predetermined therapeutic approaches but to be flexible and create strategies based on ‘the client’s cultural background, worldview, and lived experiences’ [32] (p. 7).

These theoretical perspectives fed into the emergence of *liberation psychology* as an interdisciplinary framework, which focuses on individuals’ lived experiences but considers mental well-being in its historical and sociopolitical contexts and aims to reveal and challenge systematic and sociocultural injustice, inequalities, discrimination, suppression, and violence that impact people’s mental health [34] (p. 3). Liberation therapists adopt a participatory, equalitarian, and flexible therapeutic approach to build a safe and trustable therapeutic alliance with the client. They investigated the problem from the client’s viewpoint and planned the treatment based on the client’s expectations and assumptions [35] (p. 176).

Another relevant approach is the *trauma-informed theory*. For example, Bloom and Farragher [36] introduced a new *sanctuary model* for running a mental health care system. They argued that current mental health services, particularly in Western countries, are ‘outdated, mechanistic, and inappropriate to human health and well-being’ (p. 28). They called for making an underlying change in the current system’s approach, techniques, therapy methods, and practices based on what they call ‘*trauma theory*’, which is a ‘scientifically informed and complex biopsychosocial understanding of what goes wrong for human beings under conditions of overwhelming stress’ [36] (p. 5). This interdisciplinary perspective draws on sociological, economic, and public health studies to explain the social determinants of mental health [36] (p. 6). The trauma-informed theory also puts emphasis on democratic, participatory, and relationship-based approaches that encourage collaboration with the client as well as other therapists and staff in a given organisation regarding the quality, timing, and type of therapy [36] (p. 12). It also highlights the complexity of human traumas and thus goes beyond the formally predetermined therapeutic methods and underlines combining treatments and innovative body-based therapies [36] (pp. 8–12).

### 3.2. Cognitive Anthropology and Psychotherapy

Related to but standing critically apart from the above therapeutic approaches to migrant health is the social science and mental health literature, including such fields as the sociology of mental health and science and technology studies, which go back to the 1960s and even further [37,38]. A major line of research in the field comes from psychological and medical anthropologists (or alternatively, mental health experts drawing on cultural anthropology) who look at mental health from a cultural perspective and study the cultural aspects of diagnosis and treatment, as well as the role that culture and sociopolitical context have in mental well-being and therapeutic strategies [39].

A pioneer in the field is Arthur Kleinman [40], who as both a psychiatrist and anthropologist questioned *transcultural psychiatry*, as it emerged in the 1970s, for its lack of attention to culturally specific categories of mental illness and the effects of culture on mental well-being (p. 4). Other critics, such as Good [41], explored how anthropology could contribute to cross-cultural research on experiencing and expressing psychiatric disorders to develop cultural diagnostic criteria. Others emphasised the importance of considering wider social determinants of health (e.g., social, economic, and political structures) and individuals’ resources and constraints on mental well-being, therapy outcomes, and seeking, accessing, and adhering to treatment and not just culture [42,43].

For example, Bennett [30] maintained that, with the help of anthropology, therapists can go beyond cultural competency, cultural humility, and social justice advocacy. He suggests adopting a *cognitive anthropological perspective* to go deeper ‘in understanding the complex layers of meaning embedded within the lived experiences of culturally diverse clients’ (p. 287). Cognitive anthropology is defined as ‘the study of the relationship between human society and human thought’. This field aims to discover ‘how people in social groups conceive of and think about the objects and events that comprise their world’ [44] (p. 1). Bennett [30] discussed how cognitive anthropology could improve communication between therapists and clients and help mental health practitioners choose the most appropriate treatment in practice. Equally important, for the purposes of the current article, is that cognitive anthropology can advance multiculturalism and social justice approaches by engaging with the meaning of migrant trauma in five layers:○Engaging the term (considering the psychosocial definition of trauma (i.e., multiple meanings of trauma in different contexts and even within a certain context));○Engaging the perception (if trauma is understood as an individualised medical disorder or a collective affliction with external causes);○Engaging the treatment (adopting a medical model based on medication and CBT or culturally appropriate treatments, which are based on understanding the client’s underlying needs);○Engaging the worldview (attention to one’s beliefs, attitudes, and behaviours that give meaning to specific experiences);○Engaging the schemata (one’s underlying and basic component of knowledge which unconsciously guides their understanding).

### 3.3. The ‘Cultural Models’ Approach

Of the various approaches within cognitive anthropology, the one with the most promise for advancing the care of migrants, refugees, and asylum seekers, particularly concerning issues of trauma, is the ‘*cultural models*’ *approach*. This approach emerged as a result of cognitive anthropology’s attempt to examine the underlying structure of cultural knowledge, moving from *componential analysis* (i.e., semantics) to the *prototype theory* (i.e., cognitive linguistics) and finally the *theory of schema* (i.e., cultural, cognitive, therapeutic, and gender-based) [44,45].

In the theory of schema, the basic unit is not a culture but a cultural model or schema. ‘Cultural models’ or cultural schemas refer to shared mental structures or a knowledge organisation of a group of people with similar life experiences in certain domains or a given phenomenon, such as marriage [46], social programmes [47], and motherhood [48], which inform their memories, perceptions, and expectations [45,49]. It should be mentioned that shared experiences might go beyond spatial and temporal boundaries. Therefore, each person might have recurring experiences shared with other groups within or beyond his or her society and thus develop schemas partially similar to those of other individuals [49,50]. One’s cultural schemas (understandings and assumptions) interact with the world structures to construct cultural meanings that are the typical ‘interpretation of some type of object or event evoked in people as a result of their similar life experiences’ [49] (p. 6). In this approach, culture is ‘not some free-floating abstract entity; rather, it consists of regular occurrences in the humanly created world, in the schemas people share as a result of these, and in the interactions between these schemas and this world’ [49] (p. 7).

The ‘cultural models’ approach presupposes a specific sense of culture. Contrary to multicultural psychology, which implies that a nation or ethnic group has a static, homogenous, coherent, bounded, and distinct culture, the ‘cultural models’ approach presupposes that cultures are complex, consisting of commonalities and consistencies as well as contests, disagreements, conflicts, and changes [51] (p. 114). It recognises intra-group differences, variations, disagreements, inconsistencies, contestations, and resistance along with cross-cultural differences or intrasocietal similarities [51]. Therefore, from a ‘cultural models’ approach, while certain schemas are shared transnationally, others are specific to a particular community and not widely shared beyond it. Additionally, some schemas vary within societies, leading to internal disagreements or conflicts. This approach to culture also conveys that cultural schemas and cultural meanings could be both durable and changeable across time [49] (pp. 85–88).

A considerable number of works in medical and psychological anthropology have employed a ‘cultural models’ perspective to investigate the relationship between cultural frameworks and individuals’ interpretation and perception of illnesses. Garro [52] posited that one’s constructed narratives of a certain illness reveal the conceptualisation and schemata that influence their perception of an illness and action towards it. Individuals draw on ‘cultural models’ of ‘what is expected to happen when someone is ill’ to interpret and make sense of their experience with illnesses and make decisions about treatment (p. 787). A study on the cultural model of distress among 102 Americans shows that 26% of the respondents, who were mostly women with lower education and economic statuses, used the term ‘nerves’ to talk about depression. Also, older respondents distinguished depression as an individual problem and a matter of personal responsibility that could be avoided and controlled from nerves as physical objects in the body, which are affected by external events and situations and thus inevitable and normal responses to hardships [53].

Still, while the ‘cultural models’ approach has been widely used, it has only recently been employed to study the mental health care challenges of migrants, particularly refugees and asylum seekers [54]. Hence, we used a case study from the UK to advance the value of a ‘cultural models’ approach to improve our understanding and care of migrants’ experiences with mental health care.

## 4. Methodology

This research is a qualitative study with an ethnographic approach, employing semi-structured interviews as the data collection method. In the semi-structured interview, the researcher had a few questions and returned to them to ensure that all interviewees were surveyed regarding the same domain. To provide a list of relevant questions, informal (pilot) interviews were conducted with four adult immigrants who used mental health services in the UK, one community leader working at a charity organisation in England, and one immigrant psychologist providing counselling services for immigrants. As shown in Table 1, regarding these conversations, 14 questions about the participants’ registration with the GP, referral to mental health services, and their experiences and satisfaction with and expectations of the therapist and the treatment they received, and their suggestions and recommendations were framed.

A purposeful (or purposive) sampling strategy was used to recruit participants regarding three criteria. The respondents had to be (1) adult immigrants, refugees, or asylum seekers who had any experience with talking therapy services in the UK in the last three years; (This period of time was considered because the participants should have had a clear memory of their experience to talk about it.) (2) living in North East England or Northern Ireland but originally from Middle Eastern or African countries; and (3) able to speak English, Farsi, or Arabic (due to the skills and facilities of the researcher). Ethical approval was confirmed by the Ethics Review Committee of the Sociology Department of Durham University. Informed consent was obtained from all subjects involved in the study. Participants were recruited with the help of two charity organisations in North East England and two community groups in Northern Ireland.

In terms of the interview conducted, 35 participants (20 females and 15 males), including 5 non-refugee immigrants and 30 refugees and asylum-seekers who had used mental health services in the UK, were interviewed. All interviews were conducted online through Zoom, MS Teams, or WhatsApp, taking between 30 and 45 min.

From this sample group, the current study focused on 25 interviews with refugees and asylum seekers (11 males and 14 females), given the focus of our study being on this category of immigrants rather than migrants in general, who had at least six sessions of talking therapy provided by the NHS or some voluntary or charity organisations during the last three years. The interviewees had been living in the UK for between one and four years, using 6–24 sessions of talking therapy. Most interviewees were from Iran, followed by Afghanistan, Syria, Somalia, Yemen, Palestine, and Sudan. The interviewees had diverse demographic characteristics and socioeconomic backgrounds. The youngest participant was 20 years old, and the oldest was 56. The interviewees’ educational levels varied from primary school to the PhD level, with most having a high school diploma or an undergraduate degree.

The interviews were transcribed, anonymised, and analysed. After an initial reading and reviewing of the transcriptions, a thematic analysis strategy was used to identify the key themes of the collected data. The themes were first surveyed through the interview questions, and then other themes were searched for based on emerging findings. To discover cultural models and schemas, frequently stated and salient themes were identified. Identical themes were labelled under more general concepts. For example, two broad schemas of appropriate treatments and therapy processes with some sub-schemas were identified. In data analysis, all implicitly referred themes, explicitly stated concepts, normative expectations, and descriptive experiences were considered. After the initial (first-level) analysis and defining the main themes and patterns, interpretation (second-level analysis) was carried out [51,55,56]. In this article, the findings are presented in terms of our proposed theoretical framework. The referred respondants are sequentially labelled as A, B, C, D in the text.

### Mental Health Care in the UK

Before presenting our findings, in this part, we provide some contextual information about talking therapy services in the UK to give a sense of what the sample in this study may have received. In 2008, the NHS established the Improving Access to Psychological Therapies (IAPT) programme to improve the quality and accessibility of mental health services for people living with the most common mental problems: anxiety, and depression. IAPT was renamed to the NHS Talking Therapies for Anxiety and Depression (TTAD) programme in 2023. In 2022–2023, 1.24 million people were visited by therapists, with over 664,000 people receiving a course of treatment and others receiving assessments and advice [57]. The services propose a range of talking therapies including cognitive behavioural therapy (CBT), guided self-help, counselling, behavioural activation, interpersonal therapy (IPT), eye movement desensitisation and reprocessing (EMDR), mindfulness-based cognitive therapy (MBCT), psychodynamic psychotherapy, and couples therapy [58]. In addition to NHS services, mental health charity organisations offer talking therapies. A survey conducted on those who used TTAD services in 2022–2023 showed that 66.5% of the clients had improved, and 49.9% recovered after completing their courses of treatment [57]. However, the community mental health survey on people’s experiences using NHS community mental health services in England [59] revealed that only 40% of the respondents thought the services were ‘definitely’ enough for their needs.

## 5. Results

In the rest of this article, we reflect on our ethnographic data (*n* = 25 interviews) to delineate how a ‘cultural models’ approach could be employed in mental health treatment to obtain awareness and sensitivity to clients’ expectations, assumptions, and situations and improve the quality and outcomes of therapy. The participants of this study evaluated their experiences with talking therapy differently, varying from quite satisfied with the services to absolutely unhappy. Nevertheless, the interviews indicate that in many cases, the clients’ underlying assumptions about therapy and their expectations of therapists differed from the services they received. In this part, by reflecting on participants’ experiences with mental health care, we will provide examples of how the ‘cultural models’ approach would offer an innovative methodological approach in mental health treatment by engaging with the client’s underlying schemas and the meaning of a given experience for them.

### 5.1. A Term’s (or Experience’s) Meaning

Bennett [30] discussed how engaging the potential meanings of a given term is the first level of deep cognitive anthropological interaction in mental health practice. A specific term might have multiple meanings in a given context or different and even conflicting meanings in various contexts. Engaging the term as a contextual matter also includes engaging the perceptions around it, as in how a traumatic term is perceived and if it is regarded as an individual issue or socially constructed problem with an external source of healing (pp. 292–295).

From a ‘cultural models’ approach, meaning is ‘a person’s interpretation of an object or event’ which ‘includes an identification of it and expectations regarding it, and, often, a feeling about it and motivation to respond to it’. Meaning is constructed through the interaction of one’s mental schemas and external circumstances [49] (p. 6). This definition of meaning implies individual variations, disagreements, and resistance to dominant interpretations. For a migrant client, specific pre- or post-migration experiences or stressors might have specific traumatic meanings. One example could be the cultural norms of the clients’ society that might seem common, normal, and taken for granted to the therapist but be perceived as traumatic by the client. Participant A (female, 22 years old, refugee), feeling unhappy with her therapist, stated the following:

‘I’m from Somalia and we have something called FGM (female genital mutilation) and we have so many other things, bad things into the woman. And I shared my story with my therapist. But she said, ‘It’s normal in Somalia, something like common in Somalia’. Yes, it’s common in Somalia, but it’s a big deal and something that makes women disabled. I didn’t like it. The way she sees the FGM is something like normal and I did not like that. She said it’s really normal, it’s common, it’s like being taken for granted… I know it’s our culture, but it’s something bad in our culture and I think everyone can understand that. From what I saw from her, it was like FGM is something easy’.

In this example, the therapist did not ask the client what FGM means to her, nor did the therapist seek out the nuances of the client’s meaning schema. As a result, the therapist did not notice the plausible traumatic meaning of the experience to the client, probably because, in terms of what FGM means to the therapist, she seemed to assume that cultures are homogenous and that a common cultural tradition is viewed similarly by everyone. However, the cultural model approach presupposes that there are multiple, variant, and conflicting ‘cultural models’ that link with the world structures to produce diverse and inconsistent cultural meanings. It highlights the existence of internal diversity, disagreement, resistance, and conflicts within a society that generate diverse implications for a term (or an experience).

Moreover, from a ‘cultural models’ approach, engaging a term’s meanings and the client’s perceptions of them is not limited to understanding them in terms of the client’s cultural background, but it also includes considering terms based on the client’s unique lived experiences and current life status. To illustrate this, although we share many cultural schemas with people having the same life experiences, our schemas are still constructed through our personal experiences and circumstances [49]. Moreover, as Strauss and Quinn [49] explained, ‘what something (a word, an object, an event) means to somebody depends on exactly what they are experiencing at the moment and the interpretive framework they bring to the moment as a result of their past experiences’ (p. 6). This consideration is significant for migrant clients with specific life situations impacting their schemas and interpretations of events. Participant B (male, 36 years old, refugee) stated this point clearly:

‘When we face a problem, it does not come from a vacuum. It comes from a social context. For example, the expectations you have of yourself in your own society, and how you perceive yourself in the current society. When you are a migrant, people have stigma and attitudes towards you; and when you are unemployed too and thus less satisfied with yourself, you will feel it more. My second therapist understands it very well and has a solution to it. But the first one, I felt that because she had lots of clients, it was like a production chain for her’.

In this example, Participant B is managing the collision of his own cultural expectations with the cultural stigma of the new country in which he is living, which results in the particular meaning that ‘unemployment’ has for him. For Participant B, ‘unemployment’ had a traumatic personal meaning (not just cultural). He appreciated his second therapist, who recognised his emotional reaction to his culturally contextualised situation and then, as a result of this insight, helped him with job seeking. This meaning was produced through the interaction of his expectations, assumptions, and life status as a stigmatised refugee.

Reflecting on this point, we suggest that a ‘cultural models’ perspective provides therapists with insight into the migrant clients’ understanding and interpretations of traumatic terms or specific experiences and not just insight into their culture. The ‘meaning of one’s experiences’ as a migrant or refugee should be considered in terms of the client’s mental models (schema and expectations) and specific life circumstances. Lack of deep engagement with traumatic terms and perceptions around them could impact therapeutic alliance and therapy outcomes.

### 5.2. Culturally Appropriate Treatment: Going beyond the Worldview

Bennett [30] maintained that after considering the contextual meanings of a traumatic term and the client’s perceptions of it, the next level of a deep cognitive anthropological interaction in mental health practice is engaging the treatment and choosing an appropriate form of therapy based on the client’s needs and sociocultural worldview, which may run counter to what the therapist sees as best, given his or her own cultural background.

As explained earlier, adopting a culturally appropriate treatment is emphasised by the multiculturalism perspective too [32]. Research shows that some migrant clients believe in different forms of coping strategies and treatments to deal with traumatic memories. For instance, studies conducted on refugee youths showed that forgetting traumatic experiences and avoiding disturbing thoughts and emotions through distraction and keeping the mind busy with schooling, socialising, and other activities are their preferred coping strategies [60,61]. Although these strategies are psychologically regarded as maladaptive and may have long-term impacts such as anxiety and sleep disorders, they could be relevant strategies for asylum seekers with no stable status or security [60,61]. Other studies also revealed that traditional and religious treatments such as going to church [62,63] and informal help seeking from family and friends are common preferred treatments among migrants [64], as Participant C (male, 26 years old, asylum seeker) in our research stated:

‘I don’t believe them (therapists). She just asked me to close my eyes, dream that I was on a beach, and talk about what I was seeing. She just wanted to take me to another life. But I didn’t feel anything about it. I would rather get back to my religion instead’.

Building on this point, Bennett [30] explained that what particularly distinguishes cognitive anthropology from the multiculturalism approach is *going beyond the worldview* and engaging the schemas as the most underlying and basic structure of one’s knowledge to find the most appropriate treatment. He suggests that a culturally diverse client’s narrative of distress would reveal their schemata and cultural expectations, which might be different from Western sociocultural constructs [30].

As we explained earlier, according to our ‘cultural models’ approach, schemas are the building blocks or components of our cultural knowledge about specific domains that structure our assumptions, perceptions, and expectations of our experiences [44,45]. While worldviews are often thought to be internally consistent, our schemas might be inconsistent and change over time because they are constructed through our different life experiences. This fact also supports the existence of diverse and conflicting cultural schemas and intrasocietal variations, as Strauss [50] posited:

‘Even people living at the same time in the same area will be exposed to a variety of ideologies and experiences (arising from particularities of their family, ethnicity, gender, class, education, religion, ideological and mass media exposure, as well as lifestyle choices), each of which has distinctive psychological effects. There is no need for these experiences to be consistent with each other or limited to a spatiotemporally contiguous population’.(p. 91)

Hence, while multicultural psychology emphasises considering the client’s worldview, the ‘cultural models’ approach drives our attention to the individual’s schemas, which might be different from the dominant cultural schemas in a certain society [50]. Moreover, compared with the perception and worldview, schemas could be more implicit and unspoken [65] and require more profound engagement with the client.

### 5.3. Clients’ Schemas of the Appropriate Treatment

In what follows, by reflecting on the interviews, we propose some examples of clients’ different schemas of therapy through the ‘cultural models’ approach and how our sample clustered around different therapeutic approaches based on background and need. Our examples are organised around two major topics: *choosing the appropriate treatment* and the *therapeutic process.*

Cognitive behavioural therapy (CBT), counselling, and guided self-help are the most common talking therapies offered by the NHS and charity organisations. However, interviewees’ reflections on their experiences indicate that except for CBT, clients are not usually informed of the psychotherapy approach or technique which is employed by the therapist, the reason for choosing that method, or the significance and relevance of it to the client’s condition. Hence, one of the most important points that participants in our study explicitly and implicitly stated was that in many cases, they did not trust the treatment methods of their therapists and believed their treatments were not strategic, relevant, or helpful. The services and treatments they received did not correspond to their schemas and expectations of appropriate therapy. That is not to say that the clients always knew what therapy they needed, but this does point to a disconnect in how and why a therapeutic modality is chosen for migrants and refugees. In this part, we describe the participants’ common schemas of appropriate treatment.

#### 5.3.1. The Value of the Therapist as a Coach or Instructor

The first schema commonly held by our interviewees (and around which many clustered) was the expectation of *the therapist as a coach or instructor*. According to this schema, the clients assumed that therapy would be based on *clear advice, suggestions, and even instructions*, and the therapist would direct the client’s decisions and actions, which is quite different from the type of non-hierarchical and more supportive and informal therapy most Western clients expect and which Western therapists tend to advocate. Hence, when the migrant clients in our study did not receive direct guidance or advice, they undervalued the therapy. Participant D (female, 27 years old, refugee), reflecting on her experiences with two therapists, stated the following:

‘My sessions were not really effective. They were just like being sympathised and talking to a friend… in each session, I talked about my problems and told my therapists what exactly my problem was. I was always expecting them to tell me what to do, something like an action, something objective that I could do… But it was like chatting with a friend and a non-expert. It is like going to a room and talking and being listened to and affirmed, rather than receiving counselling and solutions’.

Participant E (female, 36 years old, refugee) expressed the same expectation when comparing her two therapists:

‘My current therapist suggests me do techniques or methods to solve the issue. So it’s quite helpful. But the first one just listened to me. This one both listens and suggests. She gives suggestions on how to overcome the issue, what the right way is to solve the problem, and how to cope with your stress. But the first one was like more towards listening’.

Relative to the ‘coaching’ schema, the interviewees also stated the expectation of being advised and guided about their life challenges, rather than just having a listening ear, as expressed by Participant F (male, 51 years old, refugee). ‘We just talked about ordinary things and my everyday activities. They didn’t give me any solutions or refer me to somewhere that I could spend my time and find some friends’. This assumption, ironically enough, is similar to a medical view of therapy as fixing people, as just one example, which much of Western psychology sees as culturally inappropriate, despite what migrant clients may actually want.

Nevertheless, not all migrants in our study saw things similarly. For example, the ‘advisory’ or ‘coaching’ schema contrasted with the expectations of some participants (a smaller cluster of people) in our study. For example, Participant G (female, 47 years old, refugee) stated, ‘I expected my therapist to listen to me, empathise with me, and talk to me to help me calm down and relieve my anxiety; and she was like this’. This variation in expectations is supported by the Healthwatch survey, where the participants stated that they were appreciative of just having a chance to openly talk about their issues and emotions [66].

#### 5.3.2. The Value of Psychodynamic Therapy

The second schema of appropriate treatment among our participants (around which a smaller group of clients clustered) was *psychodynamic therapy*. This expectation was mostly held by the clients who had some psychological knowledge and expected their therapist to deal with their deep and long-lasting mental issues from the past, but their therapists were not experts in this area, or the number of allocated sessions was not enough for such deep psychoanalysis. In such cases, the therapists simply avoided dealing with the clients’ deep-rooted mental issues, rather than justifying and clarifying their expertise or the timing constraints that kept them from performing more in-depth therapy. Consequently, the clients underestimated the therapists’ diagnoses and knowledge and concluded that they were not professional or responsible enough. This perception damaged the trust in the therapist and the therapy outcome. For example, Participant H (female, 21 years old, asylum seeker) believed that therapists could not or would not pay attention to deeper psychological problems:

‘They paid attention to my short-term problems but avoided dealing with deep and long-term issues. They focussed on smaller problems that could be solved in a short time. For example, if I told them I have low self-esteem or cannot concentrate, they could give me advice. But when I mentioned something that had been bothering me since my childhood, they did not take it seriously. I knew dealing with that problem needed more time, but they did not say it explicitly and just tried to avoid going through the details. I believe they thought of their contract timing and didn’t want to take more responsibility’.

Participant I (male, 41 years old, refugee) also interpreted his experience in the same way:

‘What my therapist said was helpful up to 50%. But I could not get to what I wanted. I couldn’t deal with the questions in my mind and the depression I had… the only thing that he suggested to me was to go to the gym and study English. He could not provide me with good counselling. He couldn’t answer my questions. He just suggested I keep myself busy with sports, studying, etc. I had committed suicide once and I still had suicidal thoughts. I couldn’t forget my past experiences and he couldn’t help me with that… he just gave me hope and encouraged me to forget about the past. But forgetting these things is too difficult. I expected him to solve my deep mental conflicts that I was struggling with, but he couldn’t. He didn’t do what he should have done for me. Maybe he was not empathiser enough or didn’t have the required knowledge.

#### 5.3.3. The Value of Practical Support

The third schema of appropriate treatment (around which many asylum seekers particularly clustered) was the expectation of *receiving practical support to improve life circumstances*. Whether wanting to work on deep-seated issues or not, many refugees and asylum seekers in our study expected their therapists to focus on their current life situations, particularly in terms of trauma, to help them manage or overcome these challenges and difficulties. However, many clients in our study found their therapists, for example, employing narrative exposure therapy (NET) without justifying this therapeutic method. Therapists seemed to take it for granted that this strategy is helpful for a traumatised migrant client, not considering that repeatedly talking about their trauma but not helping them find solutions (given they are in a new country) may be counterproductive, as Participant J (female, 21 years old, refugee) commented on her two therapists:

‘The first one was good too, but I don’t think it had a benefit for me because I think she just reminded me of my situation, and what happened to me. There was no solution. She didn’t give me any solution. She didn’t give me advice… it is like I remind you of what is your problem, and at the same time I don’t have any solution for you. So, I stopped attending sessions… I can’t go there and get no benefit. But for the new one, I use his advice. He tells me if it is wrong or right… he gives me information about the rules in this country… for example, sometimes I lose my temper and shout at my housemates. He told me that fighting is not good, and you can contact the police if you have a problem. If I do anything wrong, he gives me advice’.

This client’s comments resonate with the current literature. Studies conducted on refugees and asylum seekers have demonstrated that although these individuals are traumatised and need treatment, what mostly impacts their mental health is their living situation in the host country [15,67,68]. Our study also revealed that for many refugees and asylum seekers, practical support was regarded as the best treatment, and they expected their therapists to help them overcome their tough life situations, as Participant K (female, 21 years old, refugee) put it:

‘The support that she provided for me was really good because I could realise that she was really concerned about me… at that moment and situation, I really wanted it. I wanted something or someone to help me. For instance, once I told my second therapist that they were going to accommodate us in a new house, and we wouldn’t have internet there. She introduced me to an organisation that provided me with a sim card and internet for six months. Another time, I was subjected to a fine of £50 for not scanning my bus card because I didn’t know I should do it at the bus stop. She referred me to an organisation that helped me defend myself and explained that I didn’t know the system… she helped me as much as she could’.

Participant J and Participant B also appreciated their therapists because of the practical support they provided:

‘He is a perfect counsellor. He supports me a lot. For example, my accommodation’s window was broken, and the water came down to my room when raining. I talked to him and showed him a picture of it. He called my case worker and Migrant Help. The next day they came and fixed the window’.

‘He had clear solutions for my problems. He referred me to some organisations, and it worked for me… He referred me to a job agency for refugees. They called me and I went to meet them, and it was very helpful, even better than the job centre. They heart-fully helped me, although they had limited facilities’.

Reflecting on the above insights, it is key to recognise that although the role of a psychotherapist is traditionally defined differently from a social worker’s tasks, as was discussed, the ‘cultural models’ approach draws our attention to sociocultural, structural, and political factors affecting marginalised groups’ mental well-being. Therapists need to know how to identify and respond to barriers and know the available resources for patients and families [43,69]. The ‘cultural models’ approach, by engaging the client’s schemas and life situations, provides a more holistic and integrative approach to human traumas and encourages therapists to go beyond predetermined standardised treatments and identify relevant and helpful therapeutic methods. This perspective encourages therapists to address their clients’ schemas of appropriate therapy and explain what they could or would offer.

To repeat a point, migrant clients are not a homogenous group, and they hold various expectations for mental health treatment. According to the ‘cultural models’ approach, schemas, as the mental models constructed through one’s life, not only embody culturally shared knowledge but also reflect individual and unique expectations. Therefore, in psychotherapy, an individual’s schemas should be addressed and recognised to provide a personalised therapeutic experience and avoid cultural essentialism that regards a culture’s members as a homogenous group.

The ‘cultural models’ approach also presupposes that schemas evolve and transform over time, as they are influenced by new experiences. Therefore, it acknowledges changes in a client’s expectations over the course of treatment. Participant L (male, 32 years old, refugee) stated this change in his expectations quite well:

‘Something that surprised me was that at first sessions, I asked my therapist to tell me what to do, to advise me so that I know what I should do. But I was told that they could not tell me what to do or make a decision for me, and it was interesting for me. She said that you are the only one who can decide, and I can just listen to you… but I liked to be a bit challenged. So, in the beginning, I was not very happy with the service. But I gradually learnt that it is what it is and here, psychologists don’t tell you what to do. They may ask you some questions to stimulate your subconscious mind to find the answers… Due to our cultural background, we need, which is wrong I think, to be told what to do. But they say no, you should help yourself’.

### 5.4. Clients’ Schemas of the Therapy Process

The second major topic around which clients’ schemas revolved was their expectations of the therapy’s processes and frameworks. Although therapists are supposed to follow predetermined administrative processes and principles, clients’ expectations are based on flexibility and patience. In the case of migrant clients, one important consideration is flexibility regarding the presence of an interpreter. Although available studies frequently emphasise the significance of employing trained interpreters [70], some interviewees stated that although their English was not perfect, they would have preferred to talk to their therapists directly without an interpreter and alternatively use their phones’ translators in case they did not know the meanings of some terms, because they did not feel comfortable with having a third person in their sessions, as Participant J put it:

‘My first counsellor was also a good woman, but I couldn’t feel comfortable talking to her because the interpreter was a barrier between us… I didn’t feel comfortable that the interpreter was with us. It took a long time until the interpreter explained to the counsellor and then me, and it was boring for me… but for my second counselling, the first thing I said was that I didn’t want an interpreter… because my counsellor could understand me. And sometimes if he does not understand my English, he asks me if I mean like that. And if I don’t understand his words, he explains it to me, or I use my phone’s translator’.

Participant M (female, 28-year-old, refugee) also had similar experience:

‘My second therapist was very kind and patient. Although my English was not good, I didn’t like to have an interpreter because I live in a small town, and I do not trust interpreters. My second therapist was patient and allowed me to use my phone translator. But with the first counsellor, there was a new interpreter in each session, and they were mostly men, and it was tough for me’.

Nevertheless, some therapists are strict and emphasise that the interpreter must be present at sessions, as Participant B put it:

‘In my first therapy, I had an interpreter. After the first session, I told my therapist that I didn’t feel comfortable with the interpreter… But my therapist said no, and the interpreter must be in the session. I told her that now I am speaking in English with you. But she said no, there may be some terms that you don’t know their meanings. I said I would check those terms in my phone’s translator, but she didn’t accept it… but when I had sessions with my second therapist with no interpreter, I realised I did not need an interpreter at all. Because he doesn’t state very sophisticated psychological terms that I don’t understand, but she insisted that I should have an interpreter and I had to censor myself’.

Participants hold the expectation of flexibility with regard to other administrative processes too. One of these processes is the assessment form that needs to be filled out every session and might be boring or stressful for some patients. For instance, Participant J stated the following:

‘There is a questionnaire that they ask you in counselling every week. I hated that paper because my first counsellor with the interpreter asked them every week and I felt bored. It is what I didn’t like about the first counsellor. She just asked me if I do this and do that. Sometimes I just said something, just said yes to the questions, to get rid of questions because I hated them… but I think it depends on the way that counsellor asks them. My current counsellor also has that paper, but he gives it to me and allows me to read and answer it by myself. He says I will help you if you don’t understand questions’.

Participant J explained that because she could answer questions by herself and privately, she felt it was something for her to decide, as she could write what had happened to her in the last week. This statement indicates that patients expect their therapy sessions to be personalised based on their own interests and preferences rather than routine pre-defined rules.

These views are supported by the current literature. A study by Cocklin [71], conducted on 36 clients and therapists regarding the helpfulness of therapy, found that what participants agreed about was a primary preference for ‘having a sense of control over what is happening’ and ‘being able to talk freely’. If the therapist facilitates the client’s sense of control in therapy sessions, the client will be able to talk freely about emotive subjects, stay engaged in the treatment process, and make a therapeutic alliance with the therapist. These factors are positively associated with clients’ perceptions of therapy’s helpfulness [71]. Therefore, therapists should consider the specific individual and cultural needs of clients and try to be flexible, as some clients may not be prepared for some predetermined therapy method and process [72,73]. As Participant K stated, ‘My second therapist was a very understanding lady. They brought me the same interpreter that I had requested... The sessions went on according to my will. She paid attention to whatever I wanted…’.

Flexibility is expected regarding the therapy’s stages too. For example, some clients do not feel comfortable talking about some specific topic with others, and thus they need more time to trust their therapists. Participant A put it as follows:

‘I didn’t like the way she was asking me the questions. You know, everyone has their own story, and I need to feel comfortable to share my story with you. But at first, I didn’t feel comfortable with her. In our culture, women don’t share what happened to them. Always they’re silent… I was in Greece before, and I had experience with the psychologist there. I know they listen to you. But they listen to what I want to tell them, not what they ask me. But my therapist here asked me a lot of questions, and all questions were about my life story. I told her I was not feeling happy to share my story with you right now. I want to know you first and then maybe I would trust you. But from the first day, I can’t share the whole of my story with you. But she said, no, you can tell me, just close your eyes, and speak and tell me everything that happened to you… I expected her to wait for me till I trusted her and got ready to share what I was feeling. If you say to me, tell me and tell me and tell me, I will feel like you are someone who wants to know my story, and you are just interested in my story. But you are not interested in the way I want to tell you, the way I want to share my story… I like my psychologist to give me time, listen to me and be patient’.

This expectation was also embodied in the words of Participant *n* (female, 30 years old, asylum seeker), who talked about her therapist:

‘My sessions helped me a lot because I needed someone to talk to and share my feeling with… she listens to me when I’m feeling unhappy… She listen to me a lot, and I tell her whatever I want to share, and I’m not forced to share what I don’t want to share… She listens to me and believes me’.

Reflecting on our second topic, it seems that the clients’ expectations and assumptions of therapy often differed from the therapists’ approaches and standardised formats advocated by their Western-trained therapists. Therapists should be aware of these potential differences and try to build a therapeutic alliance by explaining their strategies and expertise to clients and being flexible and patient. A ‘cultural models’ approach would help them to consider clients’ underlying schemas.

## 6. Conclusions

The main purpose of this article was to propose a new conceptual approach to migrants’ psychotherapy and a novel methodology for uncovering the value of such an approach and what key topics are of concern. It was discussed that refugees and asylum seekers with various pre- and post-migration traumatic experiences might need different therapeutic approaches.

In response, we outlined how the ‘cultural models’ approach in cognitive anthropology can provide the necessary framework and method for making such key therapeutic decisions. In this way, it is a major advance over (but also in general concert with) the values of multiculturalism, social justice, liberation psychology, and trauma-informed frameworks. As a quick summary, the ‘cultural models’ approach accomplishes the following:It offers a holistic approach to mental health and an explicit set of tools for rigorous engagement with migrant’s expectations, assumptions, and particular life situations and needs.It highlights the social, cultural, and political contexts of traumas and democratic collaboration between the therapist and client in choosing the therapeutic method.It studies the underlying components of one’s expectations and assumptions to offer a beneficial approach to mental health treatment.It considers the meaning of a term or experience for clients regarding their schemas and life circumstances as not just part of their cultural or national background, which often leads to stereotypes and incorrect therapeutic treatment.It recognises individual differences and does not regard migrant clients as a homogeneous group with the same expectations and assumptions. For example, as this research showed, while appropriate treatment for some migrant clients is providing practical support and helping them overcome life challenges and difficulties, some others expect the therapist to conduct deep psychoanalysis. Engaging the client’s schemas would reveal these variations, and the ‘cultural models’ approach, theoretically and methodologically, would facilitate this engagement in psychotherapy.

We hope this paper has shown that refugees and asylum seekers present to Western countries a specific set of mental health challenges that require a more nuanced therapeutic awareness and approach, which goes beyond the shallow multicultural psychology or psychiatry movement and toward something much more rigorous and useful. It is less about gathering factors or figures about the migrants we care for and more about listening to and engaging them where they are at and with what they present and then going on the journey with them to find mental health and well-being in their new countries. Having said that, we do recognise that even our recommendations may vary as well, based on time in a country and length of therapy involvement, which future research could explore. We also recognise that our model (as with any model) is not sufficient, and its arguments and insights could be potentially framed similarly using other models. Still, we see our model as a major advance over current approaches.

This research suggests practical changes in mental health practice through introducing therapists to more holistic and interdisciplinary approaches, such as cultural models, which encourage more engagement with clients’ schemas and provide personalised treatment. It also calls for including these theories in the curriculum for psychological treatments to develop and improve (and critically interrogate) available theoretical approaches, such as multiculturalism.

## Figures and Tables

**Table 1 ijerph-21-00650-t001:** 14 Questions for semi-structured interview with *n* = 25 clients for the study.

Questions	
1.	How long after arrival could you register with the GP?
2.	How long after arrival were you able to access counselling?
3.	Did you face any challenges or difficulties in registering with the GP and accessing the services? If so, what were they?
4.	How were you informed of available services? Who referred you to counselling?
5.	What kind of services are you using now or did you use? Talking therapy? Medication?
6.	How often are or were you visited by your therapist?
7.	Are or were you happy with the services you received? Why?
8.	Do you think the treatment is or was effective for your mental health? Why?
9.	Do you think that your therapist understands or understood your expectations and beliefs?
10.	What do or did you expect from your therapist?
11.	Do you think your therapist treats you respectfully and respects your ethnicity and cultural background?
12.	Do you think what your therapist says or said and suggests or suggested to you is or was acceptable and understandable to you? Why?
13.	Do you think individual sessions are or were useful for your mental health? Why?
14.	What are other services or activities do you think would be useful for your mental health?

## Data Availability

In adherence with research ethics approval, M.Z. is constrained from disclosing the raw data.

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
