# Peer review of "A ‘Cultural Models’ Approach to Psychotherapy for Refugees and Asylum Seekers: A Case Study from the UK"

_ijerph, 2024, doi:10.3390/ijerph21050650_

Round 1

Reviewer 1 Report

Comments and Suggestions for Authors

Dear authors,

I appreciated the opportunity to review your manuscript. I believe this topic is quite timely and relevant, and offers a novel approach to addressing the needs of refugees and asylum seekers following resettlement. 

In reading your work, I would like to offer the following suggested revisions for your consideration:

1) In describing your sample of interviewees, some additional information would be helpful to better understand their status. Particularly, length of time since resettlement (i.e., how long have they lived in their place of resettlement?) and if possible, length of time receiving mental health treatment. I would imagine that needs and perspectives may vary based on these factors, but perhaps not.

2) Subheadings may be beneficial in describing the research findings. It seems that there are key points that are lost in this section that could be drawn out with clearer subheadings. For example, when describing the schemas of appropriate treatment, the italicized texts could serve as possible subheadings to make the results clearer. (See, for example, line 522 on page 11 - italicized text: the therapist as a coach or instructor). 

3) This is an impactful and innovative approach to mental health treatment. In the conclusion, is it possible to include a "call to action?" What is the suggested change you would like to see in mental health practice?

Author Response

Response to reviewer 1:

  • In describing your sample of interviewees, some additional information would be helpful to better understand their status. Particularly, length of time since resettlement (i.e., how long have they lived in their place of resettlement?) and if possible, length of time receiving mental health treatment. I would imagine that needs and perspectives may vary based on these factors, but perhaps not.
    • Thank you for this point. The suggested information about the sample of interviewees was added to the methodology section. See lines 345-346. We also added to our conclusion the point that our recommendations may vary for people based on time in the country and length of therapy involvement. See lines 840-845.
  • Subheadings may be beneficial in describing the research findings. It seems that there are key points that are lost in this section that could be drawn out with clearer subheadings. For example, when describing the schemas of appropriate treatment, the italicized texts could serve as possible subheadings to make the results clearer. (See, for example, line 522 on page 11 - italicized text: the therapist as a coach or instructor). 
    • Because the schemas are already presented in two subsections which are not too long, adding more subheadings may not be beneficial. We do agree, however, that subheadings for the specific section mentioned are useful and so added them.
  • This is an impactful and innovative approach to mental health treatment. In the conclusion, is it possible to include a "call to action?" What is the suggested change you would like to see in mental health practice?
    • Some suggested actions were added to the end of the conclusion. See lines 846-851.

Reviewer 2 Report

Comments and Suggestions for Authors

See attached file

Author Response

Response to Reviewer 2:

The following comments are made to increase the quality of the manuscript for publication.

  • Line 86: Elaborate on these barriers to clarify the reader.
    • Barriers were added from the literature, see lines 91-98.
  • Line 87: Keep the information flowing by linking statements. This will increase the quality of the
    • A connective term was added.
  • Line 98: The example was indicated in the previous sentence. Link this sentence to the previous
    •  
  • Line 133: The statement looks incomplete. please correct or rephrase for the reader to understand.
    • The sentence was corrected.
  • Research Findings and Reflection: the results are well presented; however, the reader suggests that results should be included under this topic for clarity.
    • Thanks for this point. We agree the headings are a bit confusing. We have changed the headings, subheadings, etc, to be more in line with the template. This section now reads as RESULTS.
  • Line 327-334: The researcher should clarify the interviews conducted to the reader.
    • We have done so. Please see lines 337-340, which explain the interviews conducted.
  • General comments: Researchers should align the manuscript to the journal guidelines.
    • We agree. We changed the headings, etc, to be in line with guidelines.

Reviewer 3 Report

Comments and Suggestions for Authors

First of all, I would like to thank you for the opportunity to review the article focusing on the issues of refugee mental illness.

The article is well-founded with important data on the topic and the basis for the option on the therapeutic model. Perhaps this part is a little long compared to the methodology and the results.

Regarding the methodology, it would be important to describe in more detail how the content analysis of the interviews was carried out: what was the process, was it focused on identifying analysis categories or something else? Explain also the how authir made the classification of respondentes: A, B, C….

Also, is importante  explain how ethical issues were ensured.

The 8 point related whit the characterization on mental health in UK must be before the methodology and not in the results. It may also be located at the beginning of the methodology, but the way it is presented is not coherent, as the reader of the article expects to see the results of the study.

When authors presenting the results, they made a theorization of the model without demonstrating evidence from the data, that is, the excerpts from the interviews. This excerpts from the interviews could be used more in the text. In fact, the cultural model is defended, but in most of the excerpts presented is critizied by refugies. What is demonstrated is that:  when the model is successful put in practice is  when the therapist expresses empathy and is able to “walk in the shoes of the refugees” from an emotional point of view, and validating their experience, which means that the model is not always totaly used.

Therefore, it would be relevant for the authors to unequivocally demonstrate the results obtained by these 25 interviewees and subsequently criticize them in light of the therapeutic model. The conclusion needs to be more critical, as the model was not visible in all cases presented, and this needs to be clarified.

Author Response

Response to reviewer 3:

  • Regarding the methodology, it would be important to describe in more detail how the content analysis of the interviews was carried out: what was the process, was it focused on identifying analysis categories or something else? Explain also the how author made the classification of respondents: A, B, C….
    • Related issues were raised by Reviewer 2, please see the additions we added to lines 337-364. We provided the information requested.
  • Also, it is important to explain how ethical issues were ensured.
    • Thank you for this point. We added this. Please see Lines 332-335
  • Point 8 related what the characterization on mental health in UK must be before the methodology and not in the results. It may also be located at the beginning of the methodology, but the way it is presented is not coherent, as the reader of the article expects to see the results of the study.
    • Thanks for this point. We reorganised our headings, so this section is part of the Methods section. Please see line 366.
  • When authors present the results, they made a theorization of the model without demonstrating evidence from the data, that is, the excerpts from the interviews. This excerpts from the interviews could be used more in the text. In fact, the cultural model is defended, but in most of the excerpts presented is critizied by refugies. What is demonstrated is that: when the model is successful put in practice is  when the therapist expresses empathy and is able to “walk in the shoes of the refugees” from an emotional point of view, and validating their experience, which means that the model is not always totaly used. Therefore, it would be relevant for the authors to unequivocally demonstrate the results obtained by these 25 interviewees and subsequently criticize them in light of the therapeutic model. The conclusion needs to be more critical, as the model was not visible in all cases presented, and this needs to be clarified.
    • Thanks for this important point. We have gone through the RESULTS section again and added quotes to fully demonstrate the model beyond just ‘walking in the shoes of the refugees’. We think this has made the paper much stronger, so thank you. Also, we added to the conclusion a short critical reflection, pointing out that our model is not a panacea, and its insights could be obtained from other framings, but we still find our approach, overall, useful.

Reviewer 4 Report

Comments and Suggestions for Authors

- The manuscript addresses a significant topic nowadays, and the data obtained is relevant to the scientific community and professionals who work with ‘refugees’ and ‘asylum seekers’.

- The abstract clearly presents the objectives, the type of study carried out, the participants and the main results.

- The introduction is supported by recent bibliographical evidence and provides sufficient background about the topic approached by the authors, although there could be greater theoretical depth on the subject. 

- It would be important to specify the following concepts in the manuscript: “refugees” and “asylum seekers”

- The methodology is described clearly and in detail. The authors explain the type of study, the data collection instrument, and the criteria for participation in the study. Nonetheless, there could have been better clarification of the  participants. The reference to 35 participants and the reason for only including data on 25 interviewees is not entirely clear. Better clarification of this aspect in the manuscript is recommended. Also, it is recommended to review the explanation of the data analysis. The categories and indicators have not been explained sufficiently for the reader to fully understand their significance and application.

- The authors explain the relevance of the ´cultural models’ approach. Also, and not neglecting the relevance of the data presented and its discussion, it would be important to go into the conclusions in greater depth, and to explain the practical implications of the results obtained.

- It is advisable to review the formatting of the table (question 1).

- There are several quotations in the manuscript accompanied by page numbers, which are not necessary. This aspect should be revised, as well as the correction of the following bibliographical references: 12, 16, 26, 27, 32, 48, and 56.

Author Response

Response to Reviewer 4:

  • It would be important to specify the following concepts in the manuscript: “refugees” and “asylum seekers.”
    • Thanks for this point. We provided definitions of refugees and asylum seekers in the opening overview of the paper so that readers are clear on what we mean by these terms. Please see lines 50-54.
  • The methodology is described clearly and in detail. The authors explain the type of study, the data collection instrument, and the criteria for participation in the study. Nonetheless, there could have been better clarification of the participants. The reference to 35 participants and the reason for only including data on 25 interviewees is not entirely clear. Better clarification of this aspect in the manuscript is recommended. Also, it is recommended to review the explanation of the data analysis. The categories and indicators have not been explained sufficiently for the reader to fully understand their significance and application.
    • We clarified why we focused on N=25 participants. Please see lines 342-343.
  • The authors explain the relevance of the ´cultural models’ approach. Also, and not neglecting the relevance of the data presented and its discussion, it would be important to go into the conclusions in greater depth, and to explain the practical implications of the results obtained.
    • A similar issue was raised by Reviewer 1. We have added these points. Please see lines 84-848.
  • It is advisable to review the formatting of the table (question 1).
    • The table is fixed. Thanks for seeing that.
  • There are several quotations in the manuscript accompanied by page numbers, which are not necessary. This aspect should be revised, as well as the correction of the following bibliographical references: 12, 16, 26, 27, 32, 48, and 56.
    • We fixed the references, thanks for catching that. We also removed reference 27.

Round 2

Reviewer 4 Report

Comments and Suggestions for Authors

The authors have made significant efforts to enhance the manuscript. 

The manuscript  can be accepted for publication in the present form.